# The Use of Polyetheretherketone (PEEK) as an Alternative Post and Core Material: Five-Year Follow-Up Report

**DOI:** 10.3390/dj10120237

**Published:** 2022-12-12

**Authors:** Ammar T. Kasem, Mahmoud Shams, João Paulo Mendes Tribst

**Affiliations:** 1Fixed Prosthodontics Department, Faculty of Dentistry, Mansoura University, Mansoura 35516, Egypt; 2Department of Oral Regenerative Medicine, Academic Centre for Dentistry Amsterdam (ACTA), Universiteit van Amsterdam and Vrije Universiteit Amsterdam, 1081 LA Amsterdam, The Netherlands

**Keywords:** endodontically treated teeth, CAD/CAM, post and core technique

## Abstract

This clinical report demonstrated the use of polyetheretherketone (PEEK) for manufacturing of custom-made post and core in weakened endodontically treated central incisors. The PEEK structure was manufactured using computer-aided design/computer-aided manufacturing (CAD/CAM). The optimal fit of this custom-made endodontic post allowed a thinner cement layer; and removed the need to manufacture a core build-up. While supplementary clinical trials and in vitro studies are needed to totally elucidate the advantages and limitations of PEEK as an option for post and core manufacturing, this case report showed that it can be promising for a predictable and simplified treatment with five years of success.

## 1. Introduction

Dentistry has undergone significant evolution since its beginning. Earlier extraction of teeth was the most common treatment modality for innumerous conditions instead of preservation. Nowadays the clinicians look forward to conservative treatment approaches, a paradigm shift that can be attributed to the advances in endodontics and restorative dentistry associated with modern restorative techniques [1,2,3].

In cases of extensive coronal tooth destruction, where extra retention is required, a post should be inserted to the root in order to retain the core and the restoration [4,5,6] The choice between using a custom-made post or a prefabricated one relies on multiple factors, such as canal configuration, remaining tooth structure and restorative technique [7,8,9] The custom-made posts are fabricated mainly from metal, zirconia, and fiber-reinforced composite [10,11,12,13,14].

The ideal elastic modulus of the endodontic post is controversial. Stiff materials can promote high stress concentrations in the post surface but will deflect less during loading, which can be better to support the coronal restorations. However, this can result in catastrophic fractures of roots in weakened areas if overloaded with occlusal forces [15,16,17]. In clinical protocols, some materials were tested to prevent the catastrophic fractures of endodontically treated teeth [18,19].

According to the literature, the optimal material for manufacturing post and core should have an elastic modulus that accommodates the normal flexural dynamics of the root [4,6,10]. Post and core materials with dentin-like biomechanical properties could also be beneficial in minimizing the risk of root fractures or debonding [20,21]. Examples of these materials are high-performance polymers, such as polyetheretherketone PEEK [22,23].

Previous studies have shown that PEEK is a promising biocompatible material that has good shock absorbing capacity, mechanical strength, and both thermal and chemical resistance [3,15]. Few studies have examined PEEK applied as post and core materials [15,22,23]. Theoretically, post and core systems fabricated with these materials may decrease the occurrence of root fractures [24,25]. In order to prove this theory, further laboratory and clinical studies are necessary.

In addition to the adequate biomaterial, the digital workflow with CAD/CAM also plays an important role in improving the manufacturing of custom-made post and core [9,13]. It allows the anatomic structure to be digitally planned and then milled or 3D-printed [4]. With the aid of CAD/CAM, time can be saved from the processing of post and core, making the process simpler and reliable [9,11]. The CAD/CAM post and core allows for the proper control of the design as well as the thickness of the cement layer [4,12]. Therefore, the objective of this case report was to describe the clinical sequence for fabricating customized one-piece PEEK post and core units using a CAD/CAM system.

## 2. Clinical Report

A 27-year-old male patient referred to the clinics of fixed prosthodontics department at Mansoura University complaining from unacceptable esthetic and non-retentive restoration of the two maxillary central incisors, that usually falls a few days after cementation (Figure 1).

After removal of the restoration, and by clinical and radiographic examinations, it was found that the remaining teeth structures had thin walls supporting wide and good obturated canals (Figure 2). Since the last root canal treatment over six years before, the teeth had remained asymptomatic. Different treatment options were discussed, including: lithium disilicate endocrowns, fiber–resin composite build-up, metal post and core and immediate implant. However, aiming for reduced cement-layer thickness and stress concentration, CAD/CAM post-and-cores were planned to be made using a low elastic modulus high-density polymer.

It was planned to make immediate implants replacing the central incisors as the roots were weakened and had wide canals. After speaking to the patient, and after clinical and radiographic examination (periapical and cone beam) it was decided to save the remaining teeth structures with non-rigid custom-made post and core systems, and then make two crowns on central incisors and two laminates on laterals to correct their length. Cone beam revealed that there was a minimum remaining dentin thickness of about 1.5 mm in the weakest portion. After explanation of all the steps, benefits, risks, and other treatment alternatives, the patient signed a written consent accepting this line of treatment.

The indirect technique [26,27] for custom-made post and core fabrication was used (Figure 3). After removal of undermined enamel and weakened teeth structure, the 2/3 of gutta-percha in each canal was removed by gates gladden (A0008; Dentsply Maillefer, Tulsa, OK, USA). The canals were tapered and shaped with Peeso reamer (Largo drills; Dentsply Maillefer, Tulsa, OK, USA) eliminating undercuts and providing space for tapered post. To check the passive fit inside the canals, two tapered plastic posts were tried before injecting the impression material.

The posts were then removed, the light body impression material (Elite HD+ light body soft; Zhermack SpA, Badia Polesine, Italy) was then injected to the canals, and the posts were reinserted again. At the same time, the putty material (Elite HD+ putty soft; Zhermack SpA, Badia Polesine, Italy) was mixed, loaded in the tray, and inserted into the patient’s mouth. The impression was then removed and inspected for accuracy and details reproduction.

To prevent the dislodgement of temporary restoration, endocrowns (Figure 4) were made gaining the retention from the internal walls and canals. Teflon tape was inserted inside the canals to act as a stopper for the temporary resin material, the tray with putty, which was taken before removal of the restoration, was loaded with self-polymerizing Bis-Acryl composite resin material (Cool Temp; Coltene Gmbh + Co. Kg, Altstätten, Switzerland) and inserted into the patient’s mouth.

CAD/CAM was used to fabricate the post and core systems. The impression was scanned with a 3D optical scanner (Ceramill map 400; Amann Girrbach AG; Koblach, Austria), the posts were then designed with (Ceramill mind software; Amann Girrbach AG; Koblach, Austria) and milled from PEEK blank (CopraPeek Rose; WhitePeaks Dental solutions GmbH & Co. KG, Wesel, Germany) in the milling machine. The used material had adequate flexural strength (Weibull of 186.6 MPa), water absorption (4.66 µg/mm^3^), and low solubility (0.1 µg/mm^3^). According to the manufacturer, it is suitable for the production of permanent and removable dental prosthesis. Its indications contemplate crowns, bridges, abutments, secondary structures for combined dental prostheses, implant-supported full crowns in the posterior region and partly removable, screw-retained structures. A previous study reported this material with 3.7 GPa of elastic modulus and Poisson’s ratio of 0.4 [28].

In addition, CopraPeek was defined as 100% PEEK with a fracture load between 2021.82 and 1698.61 N when used as cantilever framework for implant-supported restorations [29].

To overcome the reddish color of PEEK material, opaque white material (Compo.Lign; Bredent medical GmbH & Co. KG, Senden, Germany) was added to the core and the cervical portion of the post (Figure 5).

The posts were air-abraded with alumina oxide particles (50 µm; 4 bar) air pressure for 14 s by (Basic-eco sandblaster; Renfert GmbH, Hilzingen, Germany). The tip of the sandblaster was fixed and adjusted 10 mm away from the posts [30]. PEEK primer (Visio.Link primer; Bredent medical GmbH & Co. KG, Senden, Germany) was then applied to the surface of the post [31]. Self-etching dual cure resin cement (DUO-LINK universal; Bisco Ind.; Tulsa, OK, USA) was then used for cementation of the posts in the canals.

To be conservative to the remaining teeth structures, the teeth were prepared with vertical preparation [32,33,34,35] removing old finish lines with total convergence angle of 10 degree. Gingivectomy with laser (QuickLase Quickwhite Ltd, Canterbury, UK) was done to correct the gingival zenith and the two lateral incisors were prepared for laminates to correct their length. The preparation was then finished and polished with discs (Sof-Lex; 3M ESPE; Paul, MN, USA) and the final preparation is shown in (Figure 6). Lithium disilicate ingots (IPS e.max press; Ivoclar Vivadent, Schaan, Liechtenstein). were pressed to manufacture the crowns and veneers.

The restorations were evaluated clinically (Figure 7A) for fitting, surface texture and color matching before cementation (Figure 7B–D). After that, the restorations were cleaned and dried with oil-free air before the intaglio surfaces of the crowns and veneers were etched with 9.5% hydrofluoric acid (Porcelain etchant; Bisco Ind.; Tulsa, OK, USA) for 20 s. After rinsing and drying, a silane coupling agent (Porcelain Primer; Bisco Ind.; Tulsa, OK, USA) was applied to the etched surfaces with a microbrush for 60 s.

The teeth were then isolated for cementation using a split dam technique and cotton rolls. After cleaning of the prepared teeth, the enamel was etched with 37.5% phosphoric acid (Gel etchant; Kerr Dent.; CA, USA) for 20 s. The teeth were then rinsed thoroughly for 10 s and dried before application of bonding resin (All Bond Universal; Bisco Ind.; Tulsa, OK, USA) to the entire prepared teeth surfaces. The intaglio surfaces of the crowns were loaded with dual cure resin luting cement (Duo-link Universal; Bisco Ind.; Tulsa, OK, USA) and bonded to the teeth, while the laminates were bonded with light cure resin cement (Choice 2; Bisco Ind.; Tulsa, OK, USA).

The excess cement was removed, the occlusion was checked, and finally the teeth were polished to achieve the definitive cemented ceramic restorations as shown in (Figure 8A,B). The patient was evaluated clinically and radiographically at a 60-month follow-up appointment (Figure 8C,D), and he expressed satisfaction with the results.

## 3. Discussion

In the contemporary dental literature, a variety of materials have been tested as post and core systems to reduce the catastrophic fracture of the remaining tooth structure, particularly those with close biomechanical behaviors to dentin [4,6,10]. In addition, it was reported that CAD/CAM technology provides homogeneous and standard blocks of materials manufactured for an industrially regulated milling operation [1,2,3]. Although previous studies assessed CAD/CAM post and core systems, only a few of them suggested using PEEK for post and core manufacturing [34,35,36].

A previous case report applied a similar method to produce one-piece CAD/CAM post and core using glass-fiber reinforced composite. According to the authors, the position of the CAD/CAM disc can considerably affect the mechanical behavior, failure mode, and surface roughness of the post and core [4]. This is also an important parameter when using PEEK, since it also contains fibers on its structure, mechanically responding with an anisotropic behavior.

Another in vitro study evaluating prefabricated post systems reported that a possible homogenous arrangement of fibers inside the root canal associated with improved adaptation of the post would promotes a better bonding strength behavior [5]. The present case report demonstrated that PEEK can be used to manufacture a well-adapted post with perfectly aligned fibers, justifying the success during the follow-up period and corroborating the previous statement.

An additional benefit of a monoblock-milled post and core is the removal of the post–core build-up interface, reducing the problems of gaps and voids [4]. When following the root canal anatomy, the posts also act as plungers during the luting procedure, leading to more contact between the cement-post setup and the dentin, therefore improving the final retention [4].

Moreover, a finite element study demonstrated that flexible material generates lower stress concentration at the cement layer between the post and dentin, compared with a post-and-core system made with stiffer materials [11]. Regarding the stress distribution profile, PEEK post and core (due to its low elastic modulus) show the lower stress distribution profile along the root mid-line in comparison with metal or glass-fiber posts [22]. However, another in vitro study suggested that the low modulus of elasticity of the PEEK post and core could produce more strain at the cervical margin under certain loads, resulting in micro-leakage due to more stress in the adhesive interface [14]. Despite this concern, deboning was not observed in the present case report; probably because the post and core adhesive interface is also protected by the cemented crown with a ferrule effect on it.

Another reason for the absence of bonding problems was that the custom-made post and core was used to allow an even cement layer following the root canal configuration. It was reported that a thinner cement layer reduces the polymerization shrinkage stress, increasing therefore the final bond strength [37]. Additionally, multiple factors can affect the fracture strength of prostheses, including the force of applied load, cement type and technique as well as the elastic modulus of the structure [38].

A comparative parametric inspection found that that heat-pressed PEEK post and core restorations have higher accuracy when compared to the CAD-CAM technique. The authors claim that new studies are needed to establish PEEK as a suitable, alternative material to an esthetic and customized post and core system [14]. Therefore, this case report can complement the authors’ suggestions, showing that the use of PEEK is a suitable material also for CAD/CAM technique since any misfit between the post and root canal was filled with cement.

According to the literature, the bond strength from PEEK (when sandblasted) ranges from 9.59 ± 1.58 MPa until 18.76 ± 1.97 MPa depending on the luting agent that was used [39]. In addition, a recent investigation compared PEEK and fiberglass intra-radicular posts bond strength, and concluded that milled PEEK posts seem to be a good clinical option, but they require improvement of CAD-CAM technology and advances in their adhesion [40]. According to the authors, milled posts are promising and can reduce clinical time for rehabilitation of extensively destroyed teeth [40]. The present case report corroborates with that, showing that a custom-made post can be easily made with CAD/CAM to successfully restore central incisors using PEEK as the restorative material.

Similar mechanical behavior was observed when comparing fracture resistance and failure mode of PEEK and nano-ceramic composite to manufacture CAD/CAM post and core [15]. According to the authors, a higher incidence of post-cement debonding was observed in PEEK group and the residual cement was left inside the root canal. They explained that PEEK has an inert surface, making the luting procedure a challenging clinical step. However, the authors also mentioned that they applied the load without cementing a crown on the specimen, making the load incidence on the edge of the core. This may have influenced the bonding performance of the post and cores [15].

When this clinical case was finished, the most up-to-date way to improve the aesthetics outcomes was with the use of opaque resin around the PEEK structure. However, due to the rapid development in dental materials, new blocks are available in different shades, improving the esthetic outcomes, particularly significant for anterior restorations [15].

## 4. Conclusions

Based on the exposed and in the documented follow-up, until further clinical studies with long-term follow-ups on PEEK post and core are available, the use of this material with a similar elastic modulus to that of dentin might provide acceptable results for CAD/CAM post and core manufacturing.

## Figures and Tables

**Figure 1 dentistry-10-00237-f001:**
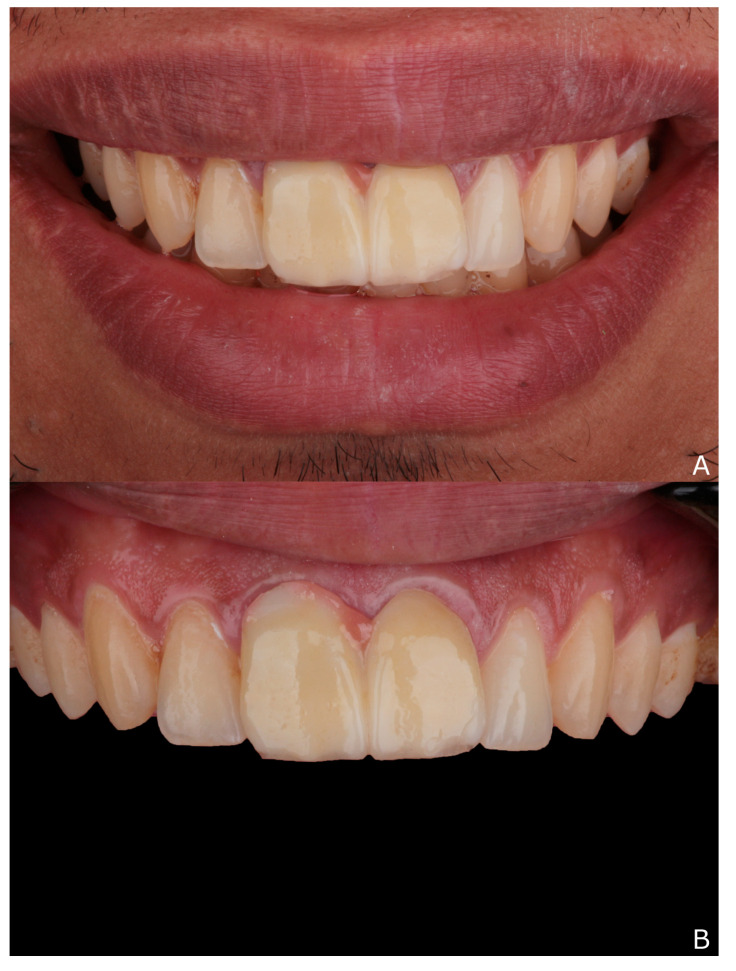
(**A**,**B**), Initial situation of two maxillary central incisors with old connected metal ceramic crowns.

**Figure 2 dentistry-10-00237-f002:**
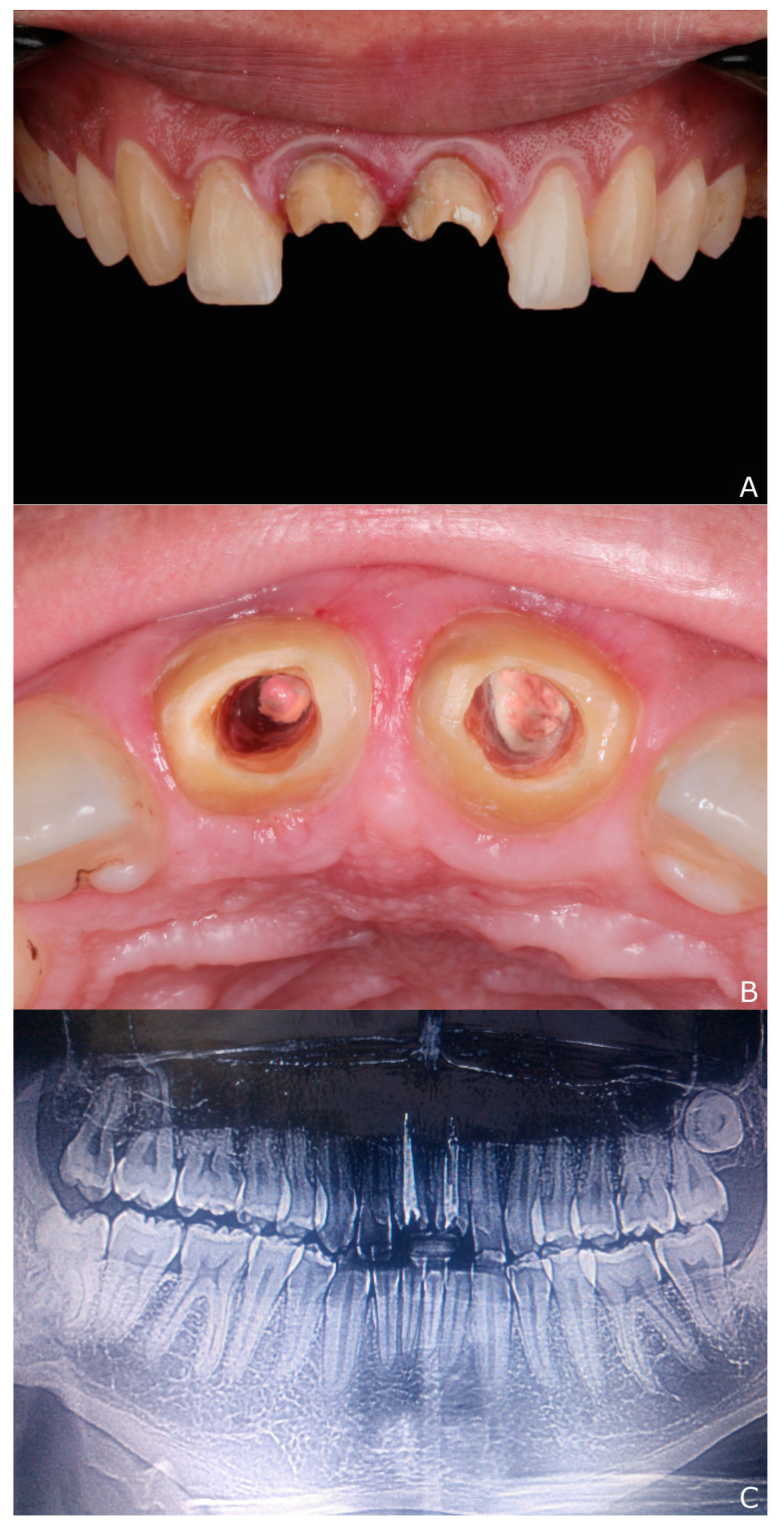
After removal of the restoration: (**A**) the remaining maxillary central incisors; (**B**) wide canals surrounded by thin-walled roots; and (**C**) radiographic examination reveals the thin remaining walls.

**Figure 3 dentistry-10-00237-f003:**
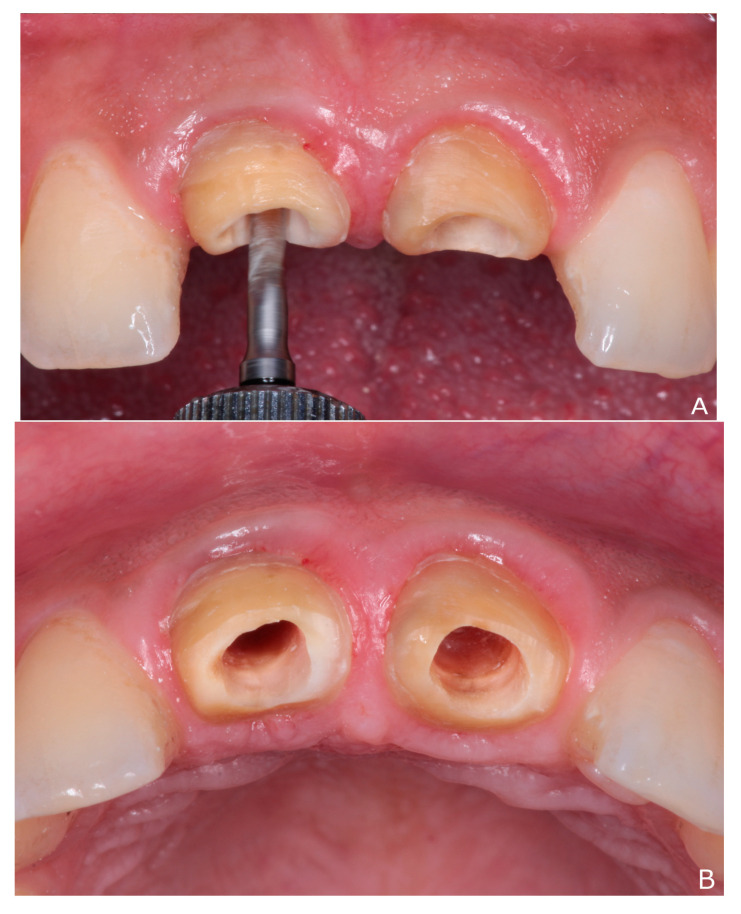
Steps of custom-made post impression: (**A**) drilling with post drill; (**B**) the canals after drilling; (**C**) plastic posts try-in to check the passive fit inside the canals; (**D**) injection of light body impression material inside the canals; and (**E**) final impression for custom-made post and core.

**Figure 4 dentistry-10-00237-f004:**
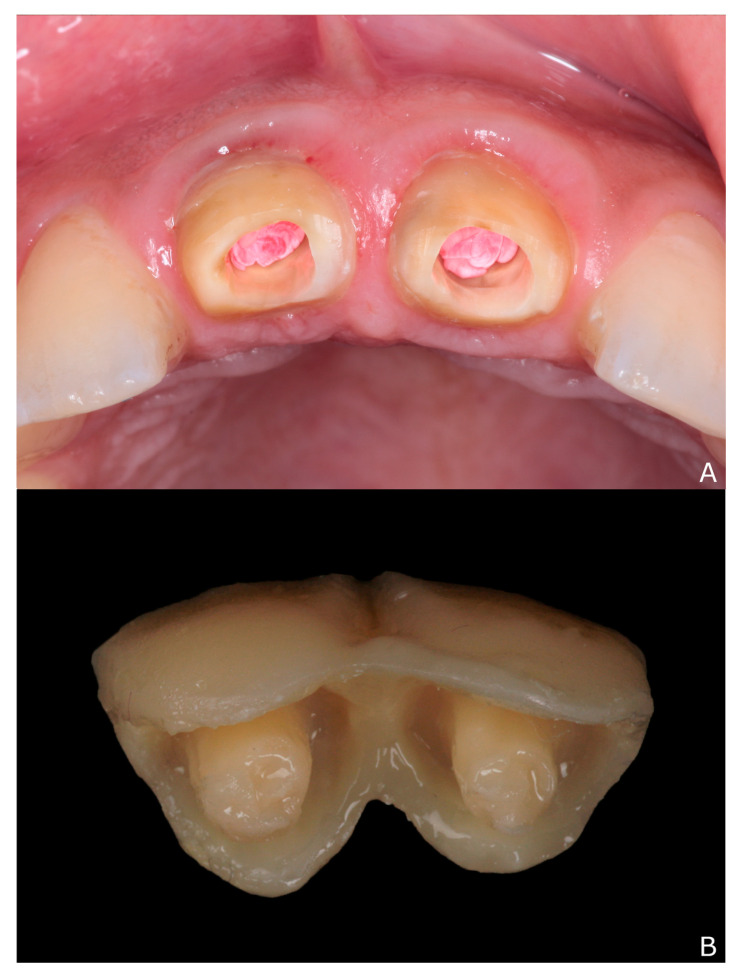
Temporary endocrown fabrication: (**A**) Teflon tape inside the canals act as stopper for the temporary resin material; and (**B**) temporary endocrowns.

**Figure 5 dentistry-10-00237-f005:**
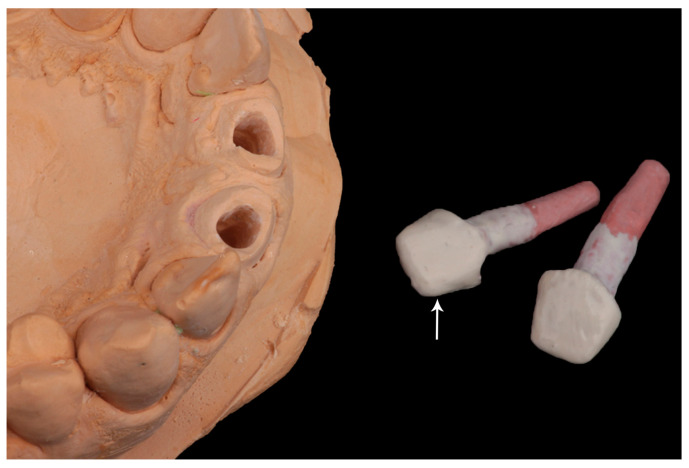
Custom-made PEEK post and core. (The arrow refers to opaque white layer to hide the reddish color of PEEK).

**Figure 6 dentistry-10-00237-f006:**
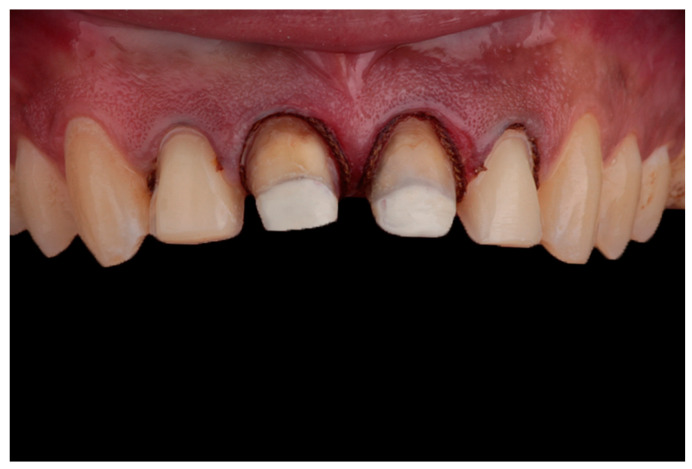
Final preparation with double retraction cord before making impression.

**Figure 7 dentistry-10-00237-f007:**
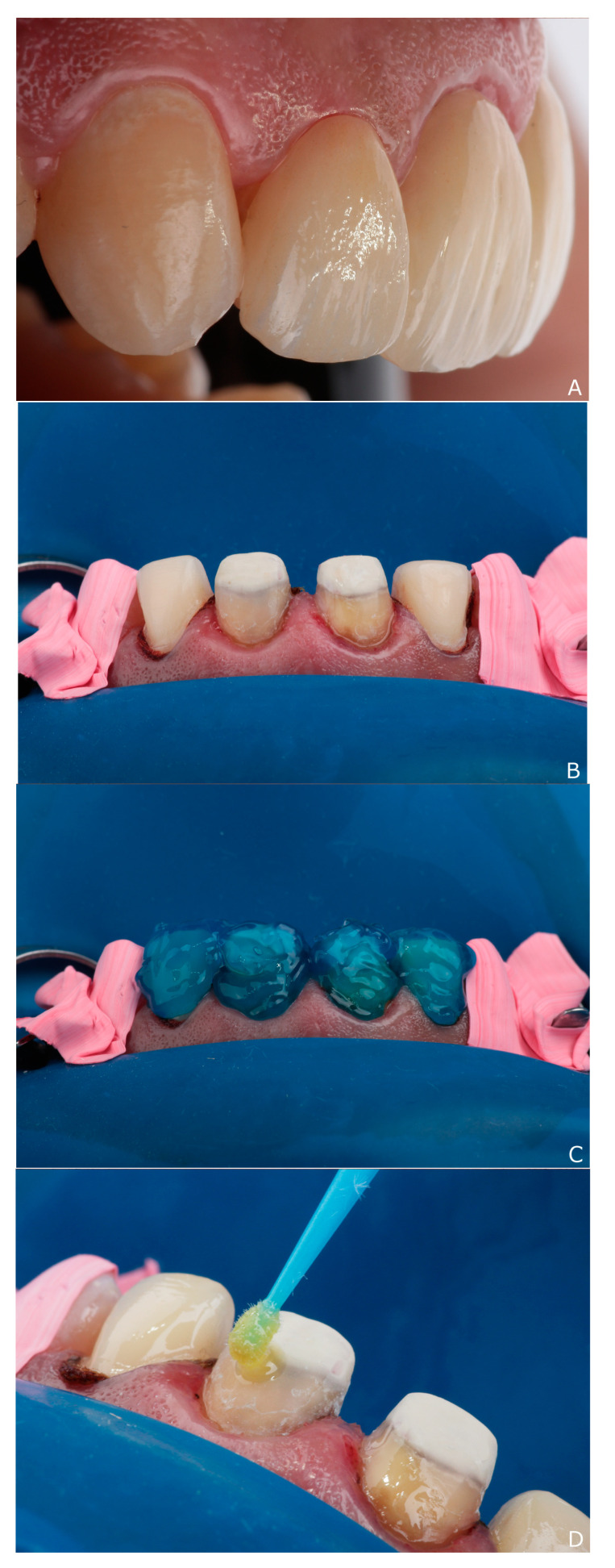
Steps of cementation: (**A**) clinical evaluation of fitting, surface texture and color matching; (**B**) isolation with split dam technique; (**C**) acid etching with 37% phosphoric acid; (**D**) bonding with universal adhesive.

**Figure 8 dentistry-10-00237-f008:**
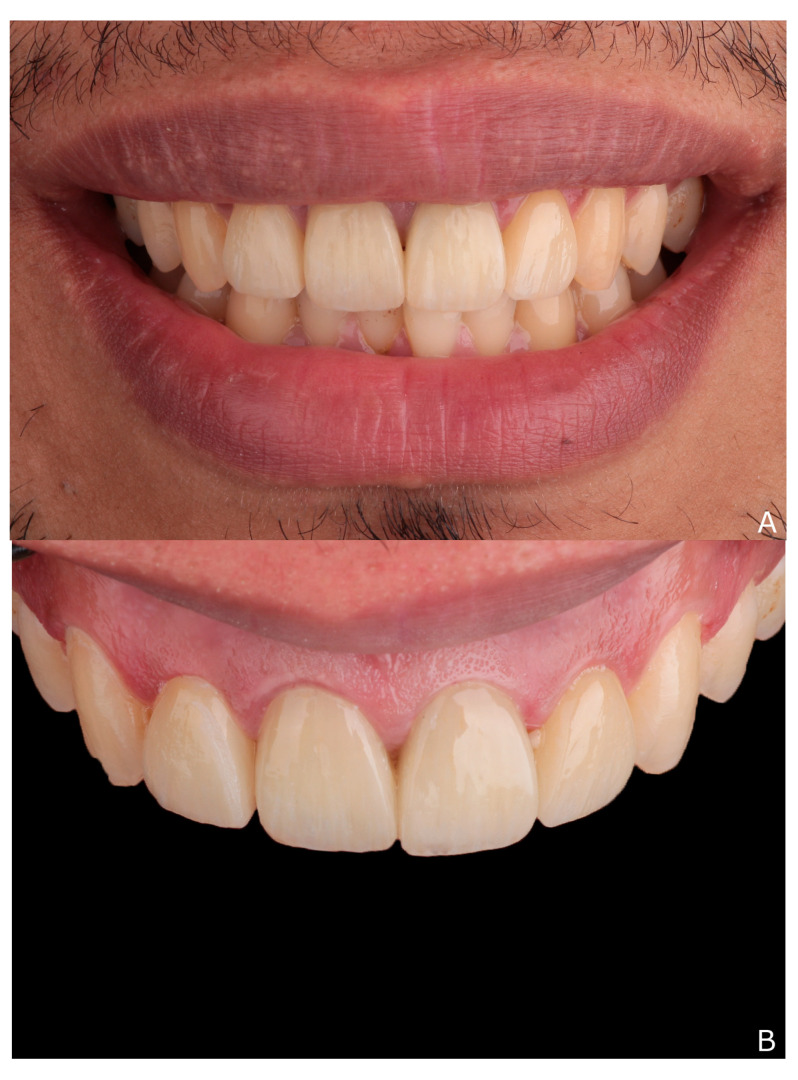
(**A**,**B**) final situation 48 h after cementation; (**C**) clinical and (**D**), radiographic examination after 60 months of clinical service.

## Data Availability

Not applicable.

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
