# Peer review of "The Use of Polyetheretherketone (PEEK) as an Alternative Post and Core Material: Five-Year Follow-Up Report"

_dentistry, 2022, doi:10.3390/dj10120237_

Round 1
Reviewer 1 Report
Dear Authors, this is an interesting study regarding tooth core reconstruction after endodontic treatment. It is a good case report, only there are a few improvements needed for it to be suitable for publication:
1) provide more information regarding the material itself - the structure, bonding to other cements/adhesive systems (can all be used?)
2) provide specific information - eg what the modulus of elasticity is exactly? What values for physical parameters were observed in previous studies?
Kind regards
Author Response
Dear Authors, this is an interesting study regarding tooth core reconstruction after endodontic treatment. It is a good case report, only there are a few improvements needed for it to be suitable for publication:
1) provide more information regarding the material itself - the structure, bonding to other cements/adhesive systems (can all be used?)
2) provide specific information - eg what the modulus of elasticity is exactly? What values for physical parameters were observed in previous studies?
Kind regards
Dear reviewer, thank you for your time and effort evaluating the present case report. We appreciate.
1) The material’s information has been inserted in the text. According to the manufacturers’ instruction the required systems is a composite primer associated with sandblasting, as used in the present case report. Other cements/adhesive systems information are not available.
2) We have added the flexural strength (186.6 MPa), water absorption (4.66 µg/mm3), solubility (-0,1 µg/mm3) according to the manufacture’s data. Additionally, the literature reports 3.7 GPa and 0.4 Poisson’s ratio for this material and fracture load between 2021.82 and 1698.61 N. The references have been added to the text.
Reviewer 2 Report
The article presents a clinical case describing a post and core for endodontically treated incisors performed with PEEK material.
The article is very well written and presents good-quality pictures.
However, the authors should make some corrections and additions.
My main concern is the use of a material that does not present a good adhesive resistance and has not been proven yet as an alternative to glass-fiber posts.
However, here I pointed out the issues to be considered:
Why did the authors use PEEK instead of Glass-fiber post CAD/CAM, since the studies show lower adhesive resistance for the PEEK material?
Page 4, line 85: What was all the alternative treatment planning for the case, except the extraction?
Page 6, line 121: Can the opaque material affect the adhesion of the core to the crown?
Page 3: The authors should present the Periapical radiography, not the panoramic one since it is impossible to plan an intraradicular post without the periapical X-Ray.
Author Response
Reviewer 2:
The article presents a clinical case describing a post and core for endodontically treated incisors performed with PEEK material. The article is very well written and presents good-quality pictures. However, the authors should make some corrections and additions.
Dear reviewer, thank you for your valuable comments and suggestions to improve the present case report. We appreciate.
My main concern is the use of a material that does not present a good adhesive resistance and has not been proven yet as an alternative to glass-fiber posts. However, here I pointed out the issues to be considered:
Why did the authors use PEEK instead of Glass-fiber post CAD/CAM, since the studies show lower adhesive resistance for the PEEK material?
Indeed, in-vitro bond strength studies have reported lower bond-strength value for PEEK, however new surface treatments and adhesive protocols have been reported to be used with this material. To assists clinicians in fully understand the clinical relevance of in-vitro results, it is important to report clinical success as the present one.
In addition, for post-and-core manufacturing, other non-adhesive dental materials such as noble alloys have been used for long-time with confirmed clinical success and high survival rate. Therefore, the present case report showed that despite the disadvantages of PEEK adhesion, it can be used with adequate outcome in at least 5-year follow-up.
About the glass-fiber CAD/CAM posts, we agree that it is nowadays a very important material to manufacturing custom-made posts and even discussed some aspects of it in the text. However, for this case, we aimed to use an alternative polymeric material with lower elastic modulus and high-density structure.
According to the literature, the bond strength from PEEK (when sandblasted) ranges from 9.59 ± 1.58 MPa until 18.76 ± 1.97 MPa depending on the luting agent that was used (Caglar et al., 2019). These values can support the clinically reported success in this case. In addition, a recent investigation compared PEEK and fiberglass intra-radicular posts bond strength and conclude that milled PEEK posts seem to be a good clinical option, but they require improvement of CAD-CAM technology and advances towards their adhesion (Monteiro).
- Caglar I, Ates SM, Yesil Duymus Z. An In Vitro Evaluation of the Effect of Various Adhesives and Surface Treatments on Bond Strength of Resin Cement to Polyetheretherketone. J Prosthodont. 2019; 28: 342-349.
- Monteiro LC, Pecorari VG, Gontijo IG, Marchi GM, Lima DA, Aguiar FH. PEEK and fiberglass intra-radicular posts: influence of resin cement and mechanical cycling on push-out bond strength. Clinical Oral Investigations. 2022; 26: 1-0.
Page 4, line 85: What was all the alternative treatment planning for the case, except the extraction?
The following sentence has been added to the text: “Different treatment options have been discussed, including: lithium disilicate endocrowns, fiber-resin composite build-up, metal post and core and immediate implant. However, aiming for reduced cement-layer thickness and stress concentration, CAD/CAM post-and-cores were planned to be made using low elastic modulus high-density polymer.”
Page 6, line 121: Can the opaque material affect the adhesion of the core to the crown?
The opaque material is a dual-curing composite adhesive. It has chemical compatibility with the resin cement used to cement the crown to the core.
Page 3: The authors should present the Periapical radiography, not the panoramic one since it is impossible to plan an intraradicular post without the periapical X-Ray.
The quality of periapical radiographs was not good enough to be inserted in the case report. However, it has been provided as requested to be available as supplementary material.